# The Stakeholders' Role in the Corporate Strategy Creation for the Sustainable Development of Russian Industrial Enterprises

Yuliya Y. Medvedeva [1], Roman S. Luchaninov [2], Natalia V. Poluyanova [3], Svetlana V. Semenova [2] and Ekaterina A. Alekseeva [4],*

1   Department of Marketing and Engineering Economics, Don State Technical University, 1 Gagarin Square, 344000 Rostov-on-Don, Russia; ymedvedeva@donstu.ru

2   Department of Management and Marketing, Belgorod State University, 85 Pobedy St., 308015 Belgorod, Russia; luchaninovrs@mail.ru (R.S.L.); semenova.s.v@mail.ru (S.V.S.)

3   Laboratory of Spatial Development, Belgorod State University, 85 Pobedy St., 308015 Belgorod, Russia; nvp-nir@mail.com

4   Institute of Industrial Management, Economics and Trade, Peter the Great St. Petersburg Polytechnic University, 29 Polytechnicheskaya, 195251 Saint Petersburg, Russia

\*   Correspondence: kapralova_ea@spbstu.ru

**Abstract:** The purpose of this study is to determine the interests of stakeholders and their influence on the strategic guidelines for the sustainable development of industrial enterprises in the Russian fuel and energy complex. Environmental safety is one of the components of the corporate strategy implementation for the sustainable development of industrial enterprises. The significance of this component is determined not so much by the regulatory institutional impact as by the transformation of economic agents' behavior in the market: consumers, partners, investors, competitors who are stakeholders. This study defines the composition of stakeholders for Russian industrial enterprises leading in the field of sustainable development based on the results of ESG-ratings. It also identifies their priority interests and the sustainable development goals of industrial companies. Based on the analysis of open reports on the sustainable development of Russian industrial enterprises, the article identifies the key goals of sustainable development for stakeholders of Russian industrial enterprises in the fuel and energy complex. An analysis of the mechanism for organizing interaction with stakeholders shows that for most stakeholder groups, interaction is implemented through the non-financial reporting of companies. The study concludes that a proactive approach will enable companies to identify sustainable development issues with the greatest long-term potential, and thus to create a system of preventive strategic action.

**Keywords:** sustainable development; greening; stakeholders; reporting; relationships; industrial enterprises

## 1. Introduction

Changes in the external environment of enterprise, and fierce competition due to the global economic crisis, have increased pressure on companies. Industrial enterprises are concerned with finding strategic and tactical solutions to adapt to the changing market environment, and to reduce the negative impact of the current recession. Solving this problem requires a change in attitude to strategic management organization.

The transformation of economic relations, the development of the economy's sustainable development concept, and toughened competition, actualize for industrial enterprises the transformation of approaches to strategic management organization. First of all, this applies to production systems where management efficiency largely depends on the balance of interests of participants (stakeholders), who can actively influence the production and commercial policy of the enterprise, distributing its resources in their favor (Mensah 2019).

According to the Energy Strategy of Russia, the main tasks for environmental protection in the development of the country's energy sector are all-round containment of growth (Batashova 2021), and reduction of the negative impact of extraction, production, transportation, and consumption of energy resources on the environment, climate, and human health. According to the regulator represented by public authorities, measures resulting in the fulfillment of this task include the integration of sustainable development indicators into the system of key performance indicators at the corporate level, development of non-financial reporting, improvement of the quality of reporting on sustainable development, and the introduction of international social corporate responsibility standards (Grishaeva et al. 2018). The implementation of sustainable development goals depends primarily on the initiative of large industrial companies, which account for most environmental pollution and greenhouse gas emissions (Shutaleva et al. 2020).

The starting point for transforming environmental landmarks is strategic planning (Farias et al. 2020). The shift in the attention of investors, government, and other stakeholders to sustainable companies reflects the general vector of the economic paradigm transformation in the Russian industrial sector. Environmental management is implemented in enterprises as a general management system component aimed at protecting the environment from the possible negative consequences of the production activities of enterprises (Panya et al. 2018), products, and services produced (Voinea et al. 2020). Environmental management is viewed both as a program, and as a process of implementing measures aimed at minimizing adverse environmental impacts and meeting the current and future needs of the stakeholders with whom the industrial enterprise collaborates (Sharpe et al. 2021).

Some studies have discussed the concept of sustainable enterprise resource planning systems. However, little research has focused on defining the stakeholders' role in transforming corporate strategies for the sustainable development of industrial enterprises.

The purpose of this study is: to determine the key interests of stakeholders in the field of sustainable development of Russian industrial enterprises; to define the ways of interaction between industrial enterprises and their stakeholders; and to study the peculiarities of Russian industrial enterprises' transition to sustainable development based on a proactive approach.

To develop a clear set of priorities, it is necessary to start with an analysis of what is most important to stakeholders throughout the value chain, with internal analysis and consultation with stakeholders, including clients, regulators, and non-governmental organizations. This process will enable companies to identify the sustainability issues with the greatest long-term potential, and thus to create a preventive system of strategic action, rather than a long list of vague desires (Jovičić et al. 2022). Edward Freeman proved the need to take into account the interests of stakeholders in order to develop relationships between market participants, and to adapt strategic guidelines to changing conditions (Freeman 1984; Freeman et al. 2010).

Hunt and Auster (1990), and Roome (1992) provide a classification of companies depending on the way environmental management is organized. As a result of these classifications, four approaches to the strategic management organization of corporate social responsibility are distinguished: reactive, defensive, adaptive, and proactive. Hart (1995) emphasizes that sustainable development, focused on minimizing environmental harm in the development of a firm, relies on clean technologies and requires a forward-looking vision shared by all stakeholders through the development of clean technologies. Based on Hart's concept, the following dominant environmental management strategies are identified: response, pollution prevention, and environmental leadership.

Professor Sulich considers style of management and decision-making within the framework of a pro-environmental strategic approach, and states that their orientation is determined by a group of endogenous (internal) organizational factors which influence managerial decision-making (Sołoducho-Pelc and Sulich 2020). This approach allows for changes in the organization and the evolution of management style. Expanding the theory

and methodology of sustainable management, Professor Sulich and co-authors explore certain aspects of green management development and its orientation to stakeholders' needs (Adam 2020; Malgorzata and Sulich 2020; Sulich et al. 2021). In the context of global uncertainty and volatility, based on a pro-environmental strategic management approach, companies are able to form competitive advantages by taking into account the interests of participants in economic relations (Jawahar and McLauglin 2001).

Because of past and present relationship experiences, a proactive approach is used to enhance those future experiences that provide a strategic vision of stakeholder relationships. When studying stakeholders' interests, a reactive approach is used, which derives from diagnosing the experience of relationships with stakeholders, and identifying the strengths and weaknesses of such interaction.

## 2. Materials and Methods

This study was based on the conceptual research method, which relies on the study of academic literature to find and integrate various concepts, including aspects of sustainable development, and strategic and operational management of an industrial enterprise.

Methods used at various stages of the study included use of mathematical statistics to assess the dynamics of environmental protection costs in the Russian Federation, and expert methods, as well as classifications, comparative literature review, and correlation of statistical and market research data.

The study relied on data from the official state statistics of Russia, which characterize the dynamics of Russian industrial sector development, and on annual financial reports, and reports on the sustainable development of Russian industrial enterprises; based on which, we studied the stakeholders' role using the expert method, methods of theoretical generalization, and comparative analysis.

## 3. Results

*3.1. Preconditions for Strategic Guidelines Transformation in Russian Industrial Enterprises*

Achieving sustainable development is the subject of discussion by international professional organizations and associations. According to the UNIDO report (Structural Change for Inclusive and Sustainable Industrial Development), the level of sustainable development of industrial enterprises is a key factor in economic growth in developing countries. Early development of manufacturing can open up opportunities for rapid and inclusive growth for these countries. The manufacturing sector plays an important role in growth, especially when countries are at relatively low-income levels (Structural Change for Inclusive and Sustainable Industrial Development 2017). For example, the manufacturing industry has a potential for higher levels of productivity, and is able to provide accelerated productivity gains with significant technological change as compared to other sectors of the economy.

Also in the manufacturing sector, jobs are being created that offer higher wages due to the higher levels of productivity achieved. Hence, there is a link between the growth of an economy and the growth of its manufacturing sector. This relationship tends to be stronger in low- and middle-income countries than in middle- and high-income countries, as productivity and employment in manufacturing, relative to other sectors, are expected to be higher at lower national income levels (Lesníková and Schmidtová 2019). Industrial development can contribute to the development of more socially significant and technologically complex industries. This structural change ensures sustainable and accelerated industrial development even after the loss of the labor cost advantage (Baker 2006).

In Russia, 2020 saw 450.5 billion rubles worth of shipped goods of own production, and of works and services performed on their own by industrial enterprises, including by types of activity: extractive industries—1.7 billion rubles; processing industries—393.6 billion rubles; provision of electricity, gas, and steam; air conditioning—44.2 billion rubles; water supply; water disposal; organization of waste collection and disposal; activities to eliminate pollution—11.0 billion rubles. In 2020, the growth rate of shipped goods of own production, works, and services performed by types of economic activity in the industry

increased by 102.9% against the corresponding period of the previous year, including: 158.8% for mining enterprises, and 102.2% for manufacturing enterprises.

The sustainable development of Russian fuel and energy complex enterprises must address their insufficient energy efficiency and environmental safety, along with the systemic importance of this sector for the national economy. Consequently, it is not possible to reduce the negative impact of these enterprises on the environment by reducing sales volumes, which means that greenifying industrial enterprises requires the efforts of all stakeholders (Medvedeva et al. 2021).

The dynamics of current costs for environmental protection in Russia are unstable, and the volumes of these costs cannot level the environmental and social risks of business (Figure 1).

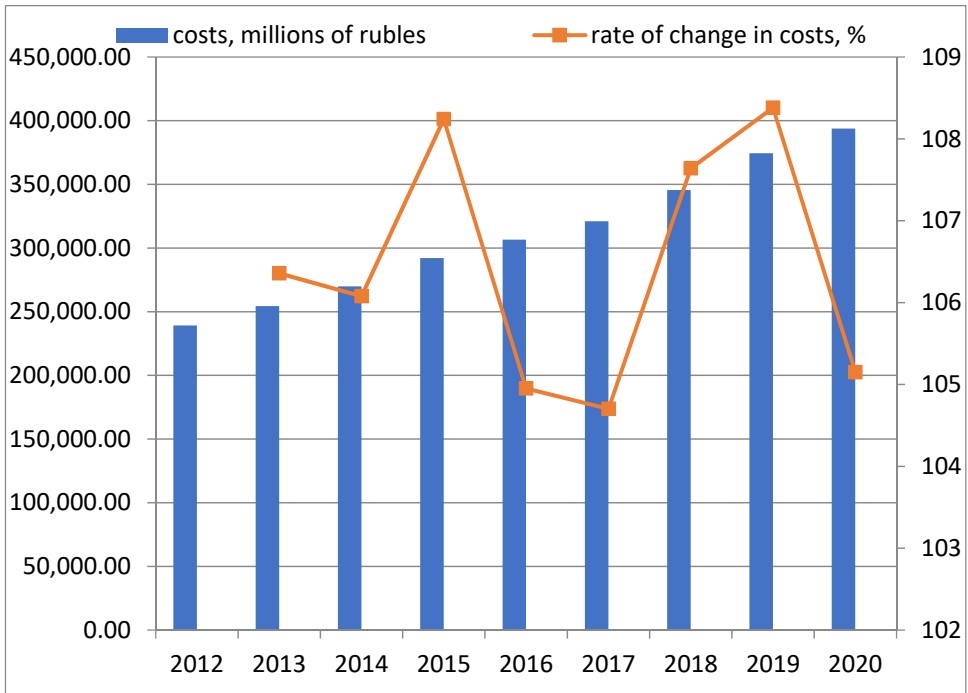

**Figure 1.** Current (operating) costs of environmental protection in the Russian Federation, and the rate of their change (*Russia in Figures* 2021).

Despite the turbulence of the national and global economy, the index of industrial production in Russia has been showing annual growth since 2016. The volumes of shipped goods of own production are increasing year by year, which entails a more intense impact on the environment. One of the important tasks facing the Russian fuel and energy complex is to ensure the rational and environmentally responsible use of energy and energy resources (Pererva et al. 2021). Reducing the level of anthropogenic impact on the environment, and increasing energy efficiency and resource conservation of production, will allow the transition of the fuel and energy complex to the principles of the best available technologies (BAT), provided for by the Federal Law of 21 July 2014 No. 219-FZ on the Introduction of Amendments into the Federal Law on Environmental Protection and into Individual Legislative Acts of the Russian Federation. However, institutional regulation currently provides neither industrial greening nor a systematic transition to the concept of sustainable development.

Stakeholders increasingly expect their company to take into account both the environmental and social implications of its decisions. Thus, businesses are forced to integrate these areas into corporate goals and strategies. Market mechanisms for transforming environmental benchmarks are currently being formed and improved, to ensure sustainable development of industrial enterprises: ESG investments intensify the competition of na-

tional enterprises for the financial attention of stakeholders with enterprises of foreign developed stock markets (Magalhães 2001; Harris 2001).

Focus on sustainable development in Russia is increasing not only at the state level, but also at the corporate level, which is a global trend. Fuel and energy companies' activities are associated with high risks of sustainable development, which requires information disclosure on measures aimed at reducing social and environmental hazards in order to increase investment attractiveness. Russian fuel and energy companies are actively involved in this process, as evidenced by the growth in the number of enterprises whose non-financial reports are publicly certified (Chkalova et al. 2019; Đordevic et al. 2021). Expecting to reduce the negative impact on the environment, industry consumers and creditors give preference to companies that cause less environmental damage and strive to green their activities (Krasyuk et al. 2019). To make a decision, the end consumer, credit institutions, and investors need to have objective information about the degree of openness of a particular company in regard to its environmental responsibility (Giddings et al. 2002).

In Russia, the practice of public assurance is just developing, which is illustrated by the increase in the number of certified reports. This trend confirms the interest of society and market participants in the non-financial intentions and practices of industrial enterprises (Table 1).

**Table 1.** Dynamics of the non-financial reports number for the fuel and energy complex that received public assurance (data source is the Russian Union of Industrialists and Entrepreneurs: https://rspp. ru/activity/social/advice/ (accessed on 3 December 2021)). ● indicates the presence of non-financial reporting in the designated year.

| Company | 2012 | 2013 | 2014 | 2015 | 2016 | 2017 | 2018 | 2019 | 2020 | 2021 |
|---|---|---|---|---|---|---|---|---|---|---|
| PJSC ANK Bashneft | | | ● | ● | ● | | | | | |
| PJSC Gazprom | | | | | | | ● | ● | ● | ● |
| PJSC Gazprom Neft | | ● | ● | ● | ● | ● | ● | ● | | |
| JSC Zarubezhneft | | ● | ● | ● | ● | ● | ● | ● | ● | ● |
| Engineering division of Rosatom state Corporation | | | | | ● | ● | ● | ● | | |
| JSC Rosenergoatom | | | | | ● | ● | | | | |
| PJSC Lukoil | | ● | | ● | | ● | ● | ● | ● | ● |
| PJSC MOESK | | ● | | | | | | | | |
| JSC MCC EuroChem | | | | ● | ● | | | | | |
| PJSC NK Rosneft | ● | | | | | | | | | |
| PJSC RusHydro | ● | ● | ● | ● | ● | ● | ● | ● | ● | ● |
| SABmiller RUS LLC | | | | | | | | | | |
| Sakhalin Energy | ● | ● | ● | ● | ● | ● | ● | ● | ● | ● |
| JSC SUEK | | | ● | | ● | | ● | | ● | |
| JSC TVEL | | | | | | ● | ● | ● | | |
| PJSC Tatneft | | ● | ● | | | | | ● | ● | |
| JSC TNK-BP | | | | | | | | | | |
| PJSC Transneft | | | | | | | | ● | ● | ● |
| EVRAZ PLC | | | | | | | ● | ● | ● | ● |
| PJSC Inter RAO UES | | | | | | | | | | ● |

As the number and variety of ESG assessments and indices grows, so does the cost of information on corporate sustainability—incompleteness and incompatibility of data remains a key issue. Public assurance consists in the recognition of the company's non-financial reporting based on the importance and completeness of the disclosed information in accordance with the principles of responsible business practice (Bespalko et al. 2019).

Public assurance of non-financial reports is a method of impartial and objective confirmation of information disclosed by companies about the implementation of the responsible business principles in corporate strategies. Reporting is reviewed by reputable specialists in corporate responsibility and non-financial reporting, who have proven themselves in

the expert and business community, which contributes to increasing public confidence in the company.

### 3.2. ESG Investment as an Indicator of the Industry's Sustainable Development

The stakeholders' composition and role, in creating the ecological vector of the strategic development for Russian industrial sector enterprises, wee investigated on the basis of contextual analysis. As a source of information, the reports of industrial enterprises included in the Russian ESG indices of investments were scrutinized and appeared to illustrate the environmental adaptation of the following Russian enterprises: PJSC Gazprom, PJSC Inter RAO, PJSC Lukoil, PJSC NK Rosneft, PJSC Sibur Holding, Evraz, PJSC Novatek, GK Rosatom, PJSC Rosseti, PJSC RusHydro, Sakhalin Energy, JSC SUEK, PJSC Tatneft, PJSC Transneft, PJSC FGC UES.

ESG indices were developed in 2019 by the Russian Union of Industrialists and Entrepreneurs (RUIE), which is not a government body, to assess the degree of implementation of the concept of sustainable development in the activities of Russian companies: "responsibility and openness" and "vector of sustainable development". These indices became the basis for the calculation by the Moscow Exchange of daily stock indices (MRRT and MRSV).

Mutual investment funds are formed on the basis of these indices in Russia. In the report of the ESG RSHB—Moscow Exchange Index RSPP Vector of sustainable development rating, such sectors as telecommunications, precious metals, energy, holdings, and non-precious metals show the greatest weight (Figure 2). The significant positions of the industrial sector are second only to the telecommunications sector, and show the values of the indices to be higher than those of the traditionally socially oriented banking sector (Indices and Ratings in the Field of Sustainable Development and Corporate Responsibility 2021).

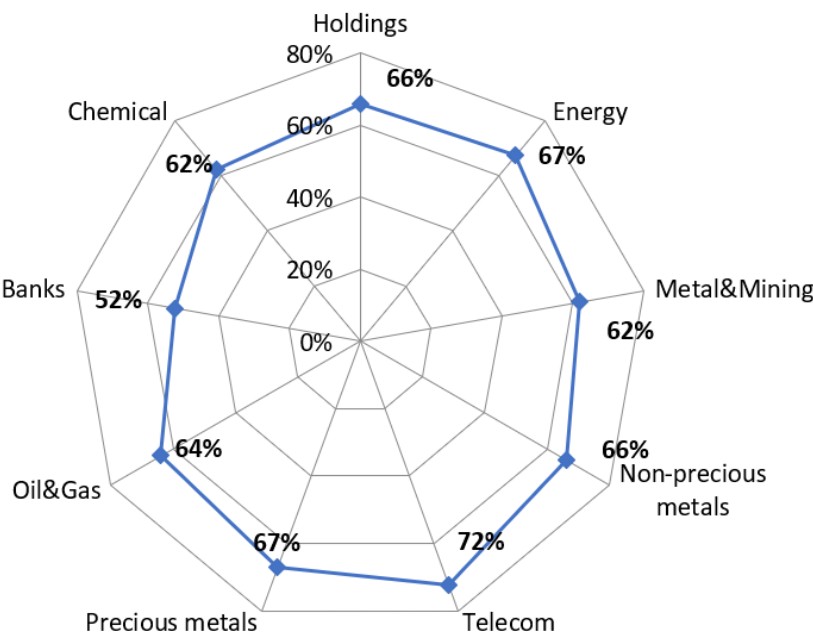

**Figure 2.** ESG RSHB Fund assessment for the Russian industrial sector (weighted), 2020 (compiled by the authors based on data from RSHB (2020).

Companies included in the RSHB—Moscow Exchange Index RSPP Vector of sustainable development fund reduce the risks associated with corporate governance. The corporate social responsibility of these companies is largely supported by a robust governance structure and transparency as well as by sound risk and performance management policies. In addition, the formation of fouling is also well mitigated in all directions. However, not all holdings have strong anti-corruption practices, which has led to several controversies related to this topic. On the other hand, companies tend to be subject to

high climate risk, with performance in all three climatic indicators also lagging behind international best practices.

*3.3. The Role of Stakeholders in Greening Corporate Strategies of Fuel and Energy Companies*

A corporate strategy for sustainable development may be considered untenable due to the fact that an enterprise is be able to fully convey the value of its strategy to recipients who have an impact on the company's activities. Consequently, long-term decisions, in regard to creating a strategy, require a transformation of approaches that can provide a synergistic effect from taking into account the stakeholders' interests within the framework of the sustainable development concept.

In the theory of stakeholders, the contingent of stakeholders is defined as part of the environment, and includes business owners, personnel or employees, business partners, investors, competitors, certain regulatory authorities, contact audiences, the media, and various public organizations. This list is not limited only to these stakeholders; the composition and quantity may vary depending on the field of activity or the market in which the enterprise operates (Babkin et al. 2020).

A stakeholder proactive approach to strategic management ensures the timely adaptation of industrial enterprises to a turbulent external environment. A proactive approach to the corporate strategy creation reduces the risks of industrial enterprises, and makes it possible to determine a strategy for environmental development (Buysse and Verbeke 2003).

Kristel Buysse and Alain Verbeke considered three proactive environmental strategies focused on taking into account the interests of stakeholders (Buysse and Verbeke 2003). The authors determined that the implementation of a significant environmental transformation requires a simultaneous increase in the efficiency of using several types of enterprise resources. At the same time, the most active and effective strategies are associated with a wide coverage, and the interests of stakeholders and their careful analysis and accounting. Environmental leadership is not ensured by regulatory institutional impact (Rossinskaya et al. 2018), but by the voluntary cooperation of enterprises with all stakeholders (Nouzha et al. 2020).

All stakeholders can be divided into two groups: internal and external ones. Internal stakeholders are important for the company as they directly influence the activities of the enterprise, and the urgency and accuracy of product delivery or service provision (Chkalova and Tikhonov 2018). Internal stakeholders include: cooperating stakeholders (founders); shareholders; top managers; different categories of staff; owners (those who have a share of the business); supplier companies (external suppliers of goods and services and subcontractors); consulting and outsourcing organizations; contact organizations (strategic partners); and the educational community (Chkalova et al. 2015).

The analysis of the sustainable development interests of stakeholder groups for Russian industrial enterprises is shown in Table 2.

The mechanism for organizing interaction with stakeholders in the majority of stakeholder groups is based on the non-financial reporting placement. As voluntary initiatives, industrial enterprises of the fuel and energy complex create socially oriented divisions that interact with consumers, the media, the scientific community, and employees. The overwhelming majority of the companies examined in the non-financial reporting enshrine activities aimed at the development of the following:

- greening of activities (reduction of consumption and improvement of water quality, reduction of polluting emissions, waste management, energy saving and energy efficiency, etc.);
- industrial safety and labor protection; the level of competencies; the system of remuneration, motivation and career development of personnel;
- social programs;
- support of local communities;
- charitable and sponsorship activities.

**Table 2.** Priority sustainable development interests of stakeholder groups for industrial enterprises of the fuel and energy complex.

| Stakeholder Group | Interests in Sustainable Development | Interaction Mechanisms |
|---|---|---|
| Employees | Social guarantees<br>Training<br>Safety | Internal communication system<br>Satisfaction surveys (polls)<br>Organization of assessment, training, and professional development<br>Motivation system |
| Investors | Increased profits and profitability by saving resources; Financial security; Accelerating return on investment; Competitive stability | Information disclosure<br>Online and offline communication on emerging issues<br>Publication of plans and indicators of their achievement |
| Partner enterprises | Reducing logistics costs<br>Terms and quality guarantees<br>Rational resource exchange<br>Energy efficiency | Contractual relationship<br>Cooperation agreements<br>Joint coordinating committees and joint working groups<br>Conferences, forums<br>Industry unions and associations<br>Prequalification of potential suppliers, contractors |
| End consumers | Environmental safety<br>Rational resource exchange<br>Energy efficiency | Contractual relationship<br>Meetings<br>Conferences, forums<br>Claims system<br>Satisfaction surveys<br>Information disclosure |
| Government and regulatory institutions | Environmental safety<br>Rational resource exchange<br>Social guarantees | Information disclosure<br>Interaction formalized by business contracts, agreements and cooperation agreements |
| Local communities, non-governmental organizations and pressure groups | Environmental safety<br>Rational resource exchange<br>Social guarantees<br>Energy efficiency | Open public hearings<br>Information centers<br>Information disclosure<br>Charity and sponsorship projects<br>Complex of educational environmental activities<br>Public opinion research<br>Providing assistance to non-profit organizations |
| Science community | Professional level of personnel<br>Environmental safety<br>Rational resource exchange<br>Energy efficiency<br>Social guarantees | Joint programs and research<br>Open public hearings<br>Information disclosure<br>Conferences, internships, industrial practice |
| Mass media | Environmental safety<br>Rational resource exchange<br>Social guarantees | Information disclosure<br>Hosting an event<br>Dealing with queries<br>Implementation of educational and cultural projects |

Thus, non-financial reporting is formed for the specific goals of the stakeholder group. The emergence of a stable vector for the formation and provision of non-financial reporting by Russian industrial enterprises of the fuel and energy complex testifies to the company's reorientation from its own interests to those of its stakeholders.

## 4. Discussion

The development of sectoral and inter-sectoral associations, as well as the development and implementation of voluntary standards for the greening of business with a view to sustainable development, is an alternative vector of institutional regulation.

The most effective motivation for sustainable development in modern Russian conditions is a direct request from stakeholders for greening business. This demand manifests itself both on the part of end consumers, in the form of an increase in demand for environmentally friendly goods and services, and on the part of investors who assess the prospects of their investments from the standpoint of green investment. Enterprises do not miss such signals from the market, which is confirmed by the annual growth of companies that publish public reports on sustainable development, and the improvement in the quality

of these reports. However, it is the regulatory impact of the state, and then that of the self-organizing branch associations, which is the basis for reorienting enterprises to the interests of stakeholders' sustainable development (Figure 3).

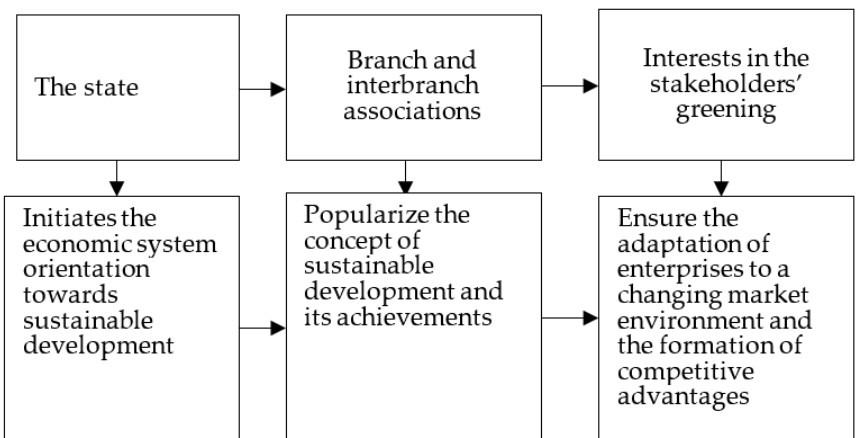

**Figure 3.** Possibilities of the subjects of economic relations to influence the greening of industrial enterprises' business.

The number of Russian enterprises that effectively implement the concept of sustainable development, in comparison with the total number of operating enterprises, is extremely insufficient to restructure the economy for green growth. Therefore, institutional regulation in the field of greening business should be present as a factor triggering these processes for the national market.

Stakeholder groups, consumers, and investors are the driving force behind the transformation of industrial enterprise strategic focus. The implementation of relationships with stakeholders is highly dependent on the quality of reporting in the area of corporate social/sustainable development. The emergence and development of ESG investment indices, assessing the transparency and completeness of disclosure of information on the sustainable development of Russian enterprises, supports this direction.

According to the Top 10 Global Consumer Trends 2020 report, an increase in sensitivity to price and value, shaping a consumer experience based on convenience and accessibility, finding a balance between online and offline formats, and caring for their own health and well-being, can be considered the main trends in consumer behavior. As the need for self-care has increased in today's environment, Russian enterprises will need to demonstrate their commitment to customer well-being by implementing innovative tools in this area. Consumers are concerned about greening consumption, but there is an imbalance in recognizing the problem and taking action to address it. For example, among Russian consumers, environmental concerns were expressed by 80% of respondents, and only 47% were ready to pay more for goods and services that did not have a negative impact on the environment. People are not ready to pay more or sacrifice comfort for the sake of forming responsible and ecological habits; rather, they are looking for those options that fit into their usual way of life. Businesses at the forefront of change are constantly looking for new ways to be sustainable, through partnerships with sustainable partners, and innovative production and processing methods.

Large and private investors see the prospects for ESG investments based on advances in environmental management (a management system in accordance with ISO 14001) and the responsible use of environmentally friendly goods (for example, cars with low fuel consumption). The management of economic and social factors in the current environment has a significant impact on the profitability, value, and share price of companies. Socially responsible investing is increasingly becoming the norm in investment as more investors are interested in clean and green products; at the same time, standards for such investment are formed, and concerns about the performance of sustainable assets are leveled.

It has been determined that under the sustainable development conditions for industrial enterprises in Russia, social factors in the context of responsible investing include aspects such as workers' rights, workplace safety, equality, education, personnel policy, standards for suppliers, social impact and relationships with local communities, and human rights, etc. ESG's corporate management criteria include transparency in relation to shareholder and board rights, and the amount of monetary compensation for board members, etc. However, in addition to environmental and social prospects, it is important for investors to clearly forecast sales volumes, profits, and capitalization of the company.

We understand the limitations of the present study, which may have affected the results. Reports of the largest 20 companies operating in the market have been used to analyze the priority interests, in the field of sustainable development, of stakeholder groups for Russian industrial enterprises in the fuel and energy complex. The results of generalizing the interests of stakeholder groups, therefore, are of a point and non-systemic nature. However, these are the largest enterprises in the industry, setting the general vector of promising management practices. We believe that in the future, smaller companies, focusing on industry leaders, will be able to find a way to effectively correct their strategies, based on the stakeholders' interests in the greening of business.

We also understand the limitation of the study results, which have been implemented only for Russian industrial companies in the fuel and energy complex. The development of the sustainable economy concept in Russian economic practice began much later than in many other developed countries, and sustainable development in economic sectors was initiated by the state through an institutional regulation system.

In general, research on the ways to form a corporate strategy for industrial enterprises' sustainable development is fragmentary and considers certain applied aspects of organizing strategic planning in order to green business activities. We believe that determining the Russian-specific causes and consequences, of implementing the stakeholder approach to reorienting the strategy of industrial enterprises towards the sustainable development concept, will expand the methodological basis for the transition of enterprises to sustainable development.

It should be noted that, in future, it will be of interest to study specific management measures and strategic decisions for each group of stakeholders in other sectors of the Russian economy, to determine the relationship between the development of green investments and a change in the enterprises' strategic orientations in the field of sustainable development, as well as to compare the practice of Russian companies with that in other countries. Also of practical interest is the definition of real and promising strategies for sustainable corporate development, in terms of creating competitive advantages for each segment of stakeholders, and improving the safety of products manufactured by such enterprises.

## 5. Conclusions

The study identified the key interests of stakeholders in the field of sustainable development of Russian industrial enterprises. It can be noted that environmental safety, rational resource exchange, and social guarantees are the priority interests of sustainable development for the stakeholders of the Russian industry.

The analysis of the mechanism for organizing interaction with stakeholders showed that for most groups of stakeholders, interaction is realized through the non-financial reporting of companies.

Most of the considered Russian companies enshrine in their non-financial reporting measures aimed at development: greening of activities, industrial safety and labour protection; the level of competencies, the system of remuneration, motivation and career development of personnel; social programs; support for local communities; and charitable and sponsorship activities.

The role of institutional regulation in the field of business greening is great, which is a factor that launches the process of transition to the concept of sustainable development of Russian industrial enterprises. The behavior of stakeholders, especially investors,

demonstrates a rethinking of the demand for indicators of economic growth, taking into account the economic, environmental and social aspects of sustainable development.

Thus, the analysis of the stakeholders' interests, and the assessment of their priorities, are the most important components of ensuring the sustainable development of industrial enterprises. The obtained results complement the knowledge in the field of strategic management organization at enterprises based on the stakeholder approach, as applied to the evolution of the participants' role in economic relations under the reorientation of industrial enterprises' strategic guidelines.

Analysis of the effective interaction directions for the stakeholder groups of industrial enterprises in the fuel and energy complex will allow their management to move to a proactive approach to the greening of business. This way of building relationships allows companies to identify sustainable development issues with the greatest long-term potential, and thus to create a system of preventive strategic action.

**Author Contributions:** Conceptualization, Y.Y.M., R.S.L., N.V.P., S.V.S. and E.A.A.; Formal analysis, Y.Y.M., R.S.L., N.V.P., S.V.S. and E.A.A.; Writing—original draft, Y.Y.M., R.S.L., N.V.P., S.V.S. and E.A.A. All authors contributed to the conceptualization, formal analysis, investigation, methodology, and writing and editing of the original draft. All authors have read and agreed to the published version of the manuscript.

**Funding:** The research is partially funded by the Ministry of Science and Higher Education of the Russian Federation under the strategic academic leadership program 'Priority 2030' (Agreement 075-15-2021-1333 dated 30 September 2021).

**Institutional Review Board Statement:** Not applicable.

**Informed Consent Statement:** Not applicable.

**Data Availability Statement:** The data will be made available on request from the corresponding author.

**Conflicts of Interest:** The authors declare no conflict of interest.

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
