# Peer review of "The Stakeholders’ Role in the Corporate Strategy Creation for the Sustainable Development of Russian Industrial Enterprises"

_economies, doi:10.3390/economies10050116_

Round 1

Reviewer 1 Report

  1. In Table 1 data sources should be placed in references.
  2. Please consider more references, because one of your methods indicates that you are preparing a literature review (line 86). Recommended literature from professor Sulich (https://scholar.google.pl/scholar?hl=pl&as_sdt=0%2C5&q=Sulich+Sustainable&btnG=)
  3. Can you improve the quality of Figure 1?
  4. Please indicate where statistical and mathematical methods are used
  5. Please provide limitations in the Discussions chapter
  6. Please indicate what are contributions (Discussions chapter) to sustainable development.
  7. Figure 2 can be upscaled to have more clarity.

Reviewer 2 Report

Authors did not present a clear objective of this work.

Firstly, this work is mostly the report-like work and there is no discussion of the obtained data. Furthermore, the presented results are not fully clear and coherent. In addition, the obtained results are not compared to any literature data. Furthermore, this work is very poorly presented in terms of idea demonstration. It is unclear what authors wish to present.

Also, the impact section is very generic with no specific message. There is no reflection on the obtained results and how they can be compared to literature reports. What are prons and cons of the proposed methodology? How the proposed approach compares to other available.

Reviewer 3 Report

Although the proposal deals with a interesting and topical topic on the sustainable development of Russian industrial enterprises of the fuel and energy complex, it would be desirable for the authors to bring some improvements to the study.

  1. The study has a lack of literature for the issue addressed. It could be improved by highlighting other studies that have dealt with similar topics.
  2. Tt would be desirable for the authors to highlight the contribution of the research undertaken compared to other previous research.
  3. The materials and methods are presented very vaguely. The authors could improve this section.
  4. Although the authors have created a discussion section, this evaluator considers it necessary to add a final section of conclusions, which clearly highlights the results of the study on the stakeholders’ role in the corporate strategy creation for the sustainable development of Russian industrial enterprises of the fuel and energy complex.
    Their implications should be discussed in the broadest possible context. Future research directions may also be highlighted.

Round 2

Reviewer 1 Report

The issues in the previous review have been corrected I approve it.

Author Response

Thank you!

Reviewer 2 Report

The article is relevant and timely, visualization (figures, tables) describes the results of the study quite well.

The title of the article is quite long (it is recommended to optimize), the abstract focuses more on environmental safety and other aspects of sustainable development; research objectives are not spelled out, not detailed; it is recommended to specify the conclusions based on the tasks set.

Reviewer 3 Report

Dear authors,

Given the changes made by the authors, I consider that the material meets the criteria for publication.

Author Response

Thank you!